# Changes in lipoproteins associated with lipid-lowering and antiplatelet strategies in patients with acute myocardial infarction

Zahra Lotfollahi[1], Ana P. Q. Mello[2], Francisco A. H. Fonseca[3]*, Luciene O. Machado[1], Andressa F. Mathias[1], Maria C. Izar[3], Nagila R. T. Damasceno[4], Cristiano L. P. Oliveira[1], Antônio M. F. Neto[1]

1 Complex Fluids Group, Instituto de Física, Universidade de São Paulo, São Paulo, Brazil, 2 Centro Universitário São Camilo, São Paulo, Brazil, 3 Escola Paulista de Medicina, Universidade Federal de São Paulo, São Paulo, Brazil, 4 Faculdade de Saúde Pública, Universidade de São Paulo, São Paulo, Brazil

* fahfonseca@terra.com.br

**Data Availability Statement:** All relevant data are within the article and its Supporting information files.

## Abstract

### Background

Despite lipid-lowering and antiplatelet therapy, the pattern of residual lipoproteins seems relevant to long-term cardiovascular outcomes. This study aims to evaluate the effects of combined therapies, commonly used in subjects with acute myocardial infarction, in the quality of low-density lipoprotein (LDL) particles.

### Methods

Prospective, open-label trial, included patients with acute myocardial infarction. Patients were randomized to antiplatelet treatment (ticagrelor or clopidogrel) and subsequently to lipid-lowering therapy (rosuvastatin or simvastatin/ezetimibe) and were followed up for six months. Nonlinear optical properties of LDL samples were examined by Gaussian laser beam (Z-scan) to verify the oxidative state of these lipoproteins, small angle X-ray scattering (SAXS) to analyze structural changes on these particles, dynamic light scattering (DLS) to estimate the particle size distribution, ultra violet (UV)-visible spectroscopy to evaluate the absorbance at wavelength 484 nm (typical from carotenoids), and polyacrylamide gel electrophoresis (Lipoprint) to analyze the LDL subfractions.

### Results

Simvastatin/ezetimibe with either clopidogrel or ticagrelor was associated with less oxidized LDL, and simvastatin/ezetimibe with ticagrelor to lower cholesterol content in the atherogenic subfractions of LDL, while rosuvastatin with ticagrelor was the only combination associated with increase in LDL size.

**Funding:** FF received financial support by the Research Foundation of the State of Sao Paulo - FAPESP (grant # 2012/51692-7) and by an investigator-initiated grant from AstraZeneca (ESR 14-10726). AF received grant by the National Institute of Science and Technology Complex Fluids (INCT-FCX) (grant # 428793/2016-9) and by FAPESP Thematic Project # 2016/24531-3 and 2018/07340-5. the funders had no role in study design, data collection and analysis, decision to publish, or preparation of the manuscript.

**Competing interests:** I confirm that this does not alter our adherence to PLOS ONE policies on sharing data and materials. The study design, data collection, statistical analysis, or publications were not influenced by the sponsors and are the exclusive responsibility of the investigators.

## Conclusions

The quality of LDL particles was influenced by the antiplatelet/lipid-lowering strategy, with ticagrelor being associated with the best performance with both lipid-lowering therapies. Trial registration: NCT02428374.

## Introduction

Lipid-lowering and antiplatelet therapies are universally prescribed for patients with acute coronary syndromes [1, 2]. The possibility of pharmacokinetic interactions, when sharing the same metabolic pathways, raised safety concerns regarding their efficacy, particularly for pro-drugs, such as simvastatin and clopidogrel [3, 4]. Ticagrelor, a reversible $P2Y_{12}$ inhibitor, differently to thienopyridine derivatives (e.g., clopidogrel or prasugrel) does not require metabolization for its antiplatelet activity [5]. Despite these differences, clopidogrel and ticagrelor share some interesting properties, suppressing nuclear factor kappa B (NF-kB) signaling and decreasing the release of several inflammatory cytokines [6]. Interestingly, among apolipoprotein E knockout (APOE-/-) mice, the anti-atherosclerotic effect with ticagrelor was associated with the downregulation of proprotein convertase subtilisin/kexin type 9 (PCSK9) [7]. Differences in the clearance of atherogenic lipoproteins and pharmacokinetic interactions may be associated with residual cardiovascular risk after lipid-lowering and antiplatelet therapy. In fact, atherogenic subfractions of lipoproteins have been associated with an increased risk of cardiovascular outcomes [8, 9]. Furthermore, even after effective lipid-lowering therapies, residual atherogenic lipoproteins also predict cardiovascular risk [10, 11], and small LDL size seems more vulnerable to oxidation and atherogenicity [12]. On the other hand, larger LDL particles with a greater number of antioxidants are considered less atherogenic and these particles can be identified by means of some properties of nonlinear optics [13, 14]. Thus, this study aimed to evaluate the effects of randomized antiplatelet and lipid-lowering strategies, commonly prescribed in subjects with ST-segment elevation myocardial infarction (STEMI), on the pattern of LDL particles assessed by complementary methods.

## Materials and methods

### Study population

For this open label study with blinded endpoints, consecutive patients of both sexes with their first myocardial infarction, from March 2018 to December 2019, were included as part of the BATTLE-AMI study, NCT02428374 [15], after the study has reached the planned number of participants. Fig 1 shows the number of patients assessed, enrolled and that completed the trial. All included patients were submitted to pharmacological thrombolysis in the first 6 h of STEMI and referred to Hospital Sao Paulo to perform coronary angiogram and percutaneous coronary intervention (PCI) when needed, in the first 24 h of STEMI (pharmacoinvasive strategy). Key exclusion criteria were clinical instability, use of lipid-lowering or immunosuppressant therapies, autoimmune disease, known malignancy, pregnancy, or signs of active infections. After hospital admission, these patients were randomized by cardiologists from the coronary care unit of *Escola Paulista de Medicina* 1:1 to receive ticagrelor 90 mg bid (Brilinta®, AstraZeneca) or clopidogrel 75 mg qd (Plavix®, Sanofi Aventis), and each group was also randomized 1:1 to be treated with rosuvastatin 20 mg qd (Crestor®, AstraZeneca) or simvastatin 40 mg plus ezetimibe 10 mg qd (Vytorin®, MSD) in two-by-two factorial design, using a central computerized system (battle-ami.huhsp.org.br). These doses of lipid-lowering drugs were

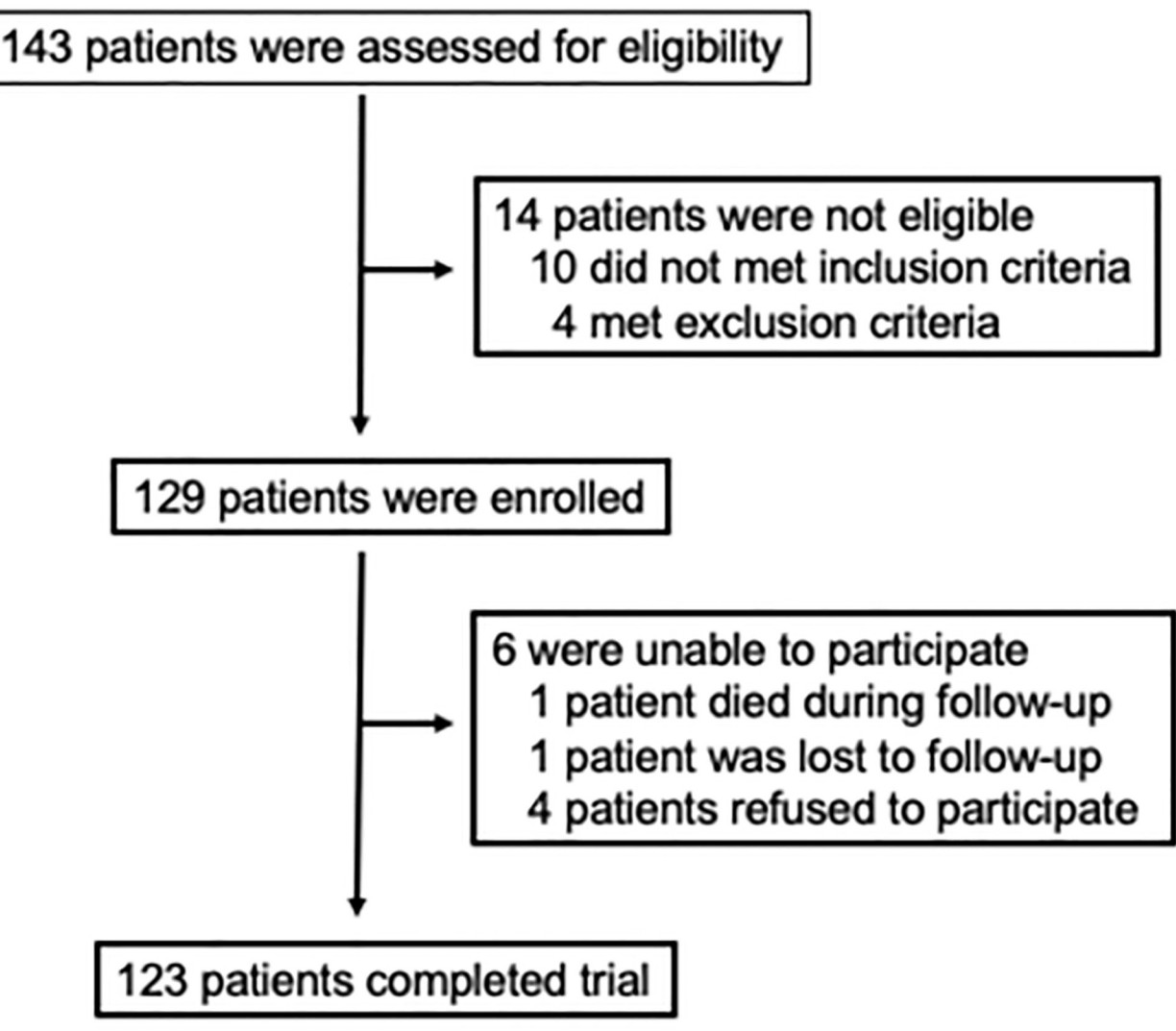

**Fig 1. Enrollment and flow of patients.** From the total of 143 subjects assessed for eligibility, 14 were excluded due to inclusion or exclusion criteria and 6 did not complete the 6 months follow-up.

chosen because of their effectiveness in reducing LDL-C by more than 50% [16]. In the last two decades a collaboration between the *Escola Paulista de Medicina* and the *Instituto de Física da Universidade de Sao Paulo* was established and this deeper analysis of the quality of lipoproteins in the BATTLE-AMI study was the result of this relationship. The study protocol was approved by the local ethics committee (*Escola Paulista de Medicina*–UNIFESP IRB 0297/ 2014; CAAE: 71652417.3.0000.5505), which follows the latest Declaration of Helsinki, and the written informed consent was provided by all subjects before their inclusion. This study was supported by the Research Foundation of the State of Sao Paulo—FAPESP (grant # 2012/ 51692-7), by an investigator-initiated grant from AstraZeneca (ESR 14–10726), by the National Institute of Science and Technology Complex Fluids (INCT-FCX) (grant # 428793/ 2016-9), and by FAPESP Thematic Project # 2016/24531-3 and 2018/07340-5. The study design, data collection, statistical analysis, or publications are not influenced by the sponsors and are exclusive responsibility of the investigators.

## Blood samples

Blood samples collected in the first day of STEMI (T0) and after six months (T6) were separated in plasma and stored at -80˚C for further analysis. All the following analyses of the quality of lipoproteins were blinded to investigators.

Low-density lipoprotein was obtained by preparative sequential ultracentrifugation equipped with a fixed-angle rotor (Hitachi Himac CP 70MX, Tokyo, Japan) as previously reported [17]. These samples were dialyzed against phosphate-buffered saline (PBS) with ethylenediaminetetraacetic acid (EDTA) to remove the salts. Routine laboratory assays were performed in the Central Laboratory of the university hospital.

## Z-scan technique

The Z-scan technique was used, as previously reported, to measure nonlinear optical properties of LDL samples [17, 18]. In this technique, LDL samples, as a weakly absorbing medium, were illuminated by a Gaussian laser beam with wavelength 532 nm. The sample converts the light energy into heat and a thermal lens is formed on it. The strength of the thermal lens depends on the medium properties such as the thermo-optic coefficient, absorption coefficient, and thermal conductivity. The phase shift θ is a dimensionless parameter that measures the strength of the thermal lens formed in the LDL sample. In the Z-scan setup, a mechanical chopper providing a square pulse (30 ms pulse width) was used to modulate the light intensity. The sample was scanned around the focal point in the z-direction. The intensity of transmitted light is measured as a function of sample z-position. More details about the Z-scan setup can be found in our previous works [17–20]. One can obtain the nonlinear phase shift θ that is related to peak to valley amplitude measured from the normalized transmittance curve. The larger peak to valley amplitude, the stronger the thermal lens strength, and the less modified LDL particle. All Z-scan experiments were carried out at the temperature of 37˚C.

## Small angle X-ray scattering

The small-angle X-ray scattering (SAXS) was applied to the structural characterization of LDL samples and to investigate structural changes in these particles [21, 22]. SAXS measurements were performed on the Xeuss 2.0 laboratory instrument (Xenocs SAS®, France), with a Cu X-ray source (wavelength $\lambda_x$ = 1.54 Å) and a Dectris Pilatus™ 300k detector. The sample to detector distance was 2.5 meters providing an experimental setup with q (modulus of the scattering vector $(4\pi \, sin\theta_x)/\lambda_x$, where $2\theta_x$ is the scattering angle) range of 0.004 < q < 0.157 Å⁻¹. The sample was placed in a homemade sample holder and the data collection was performed in a vacuum path and controlled temperature (37.0± 0.3ºC). To ensure that parameters from SAXS analysis provide information about monomeric particles, the Generalized Indirect Fourier Transformation (GIFT) method was applied. In the GIFT method the form and structure factors are decoupled. SAXS data analysis was done by a SAXS model in which, at physiological temperature, LDL is assumed as a spherical core-shell particle with a total radius R and an external monolayer of thickness ΔR. Besides, the SAXS model includes a relative electron density contrast $\mu = \Delta\rho_{core}/\Delta\rho_{shell}$ where $\Delta\rho_{core} = \rho_{core}-\rho_{buffer}$ and $\Delta\rho_{shell} = \rho_{shell}-\rho_{buffer}$. ρ is the average electron density of the respective medium.

## Dynamic light scattering

The Dynamic Light Scattering (DLS) was employed to analyze the particle-size distribution [23–25]. The DLS measurements were performed with a 90Plus particle-Size Analyzer (Brookhaven, Holtsville, NY, USA) equipped with a He–Ne laser (λ = 653 nm) and a fast photon

detector at a fixed angle of 90˚. All DLS measurements were done at the temperature of 37˚C. The DLS instrument recorded the intensity autocorrelation function, which was transformed into the particle size distribution, weighted by volume, number, and intensity of scattered light to determine particle-size information.

### UV-visible spectroscopy

The UV-visible spectroscopy is a technique for quantitative and qualitative analysis of a sample and works based on the Lambert-Beer's law. The linear absorbance spectra of the LDL samples were conducted by a UV-visible spectrophotometer with a light wavelength from 200 to 900 nm using quartz cuvettes with an optical path of 1 cm. The absorbance is computed by removing the Rayleigh scattering from the extinction spectra measured by the spectrophotometer. In this study, we investigated the absorbance values measured at wavelength 484 nm, corresponding to one of the peaks of the absorbance of carotenoids. The maximum light absorbance of other molecules present in the LDL, such as ApoB-100, cholesterol, α-tocopherol, and phospholipids, present absorption peaks between 200 to 300 nm. So, they do not contribute on forming the thermal lens in the LDL solution [17]. All UV-visible spectroscopy measurements were performed at 37˚C.

### Lipoprotein subfractions analysis

The lipoprotein subfractions analysis was assessed in *Faculdade de Saúde Pública* (*Universidade de São Paulo*–USP).

Blood samples (20 mL) were collected in the first day (T0) and six months (T6) after STEMI, and promptly stored at -80ºC until the lipoprotein analysis. The LDL subfractions were classified and measured by the Lipoprint System (Quantimetrix Corporation, CA, USA) according to the manufacturer's instructions [26, 27]. Briefly, the method is based on the separation and quantification of lipoprotein sub-fractions by non-denaturing polyacrylamide tube gel electrophoresis. To perform this procedure, 25 μL of the serum were added to the polyacrylamide gel tube and 200 μL of the dye-gel solution. The sample was homogenized, and the tubes containing the samples were photo-polymerized and subjected to the electrophoresis process. After separation of the sub-fractions, the tubes were scanned to identify each subclass [27]. The LDL-1 and LDL-2 subclasses were classified as large LDL (less atherogenic particles) while subclasses LDL-3 to LDL-7 were classified as small and dense particles (more atherogenic particles). All analyses were conducted in duplicate and coefficients of variance intra- and inter-assay were less than 15%.

### Statistical analysis

Continuous variables were expressed as median and interquartile ranges (IQR). Shapiro-Wilk test was used to verify the normality of data distribution. Categorical variables were compared by the Pearson's Chi-square test. For comparisons between groups, unpaired two sample t-test (2-tailed) or the Mann-Whitney U test, were used for variables with normal or non-Gaussian distribution, respectively. Within-group comparisons were carried out using the paired sample t-test (2-tailed), to compare groups with normal distribution or the Wilcoxon signed-rank test for non-Gaussian distribution. Comparisons between combined therapies were made by the non-parametric Kruskal-Wallis test with Dunn's post-hoc test. Correlations between continuous variables were examined by the Spearman's rank test. Convenience sample was adopted for this study. Statistical analyses were performed with the OriginPro 2020 software and IBM® SPSS® Statistics—version 23.0. Statistical significance was set at p-value < 0.05.

## Results

### Study population

The population (n = 123) was predominantly composed of overweight middle-aged men. Other characteristics of the study population are shown in Table 1. Patients included in the trial were thrombolyzed in the first 6 h of STEMI and referred to the Hospital Sao Paulo to perform coronary angiogram or coronary percutaneous intervention in the first 24 h. All these patients were in stable clinical conditions before randomization. They were accompanied in our outpatient clinic for as many visits as needed to adjust medications related to ventricular remodeling and to ensure full adherence to study protocol. The study medications were very well tolerated. During the 6 months of follow up, there was only one death and no other severe adverse event was recorded. The study design and summary of findings are showed in the graphical abstract (Fig 2).

Prospective, randomized, open label study included 123 patients with ST-segment elevation acute myocardial infarction (STEMI) and compared four lipid-lowering/antiplatelet strategies. After six months of treatment, ticagrelor with both lipid-lowering strategies, was associated with the best quality of LDL particles.

### Z-scan technique

As mentioned before, the typical result in the Z-scan experiment is a peak to valley curve. This peak to valley amplitude ($\Delta\Gamma_{pv}$) is proportional to the phase shift ($\theta$) of the thermal lens formed. The phase shift, in the context of the nonlinear optical study of the LDL solution, is a parameter that indicates how modified is the LDL. Higher values of $\theta$ indicate less modified (oxidized) LDL. Patients treated with simvastatin plus ezetimibe increased their $\theta$ values (median [IQR]) from baseline (T0) to six months (T6) (0.008 [0.004–0.019] to 0.019 [0.013–0.029], p = 0.001) while no changes in $\theta$ values were observed for those patients treated by rosuvastatin (0.015 [0.005–0.028] to 0.016 [0.011–0.026], p = 0.382) (Fig 3A).

**Table 1. Characteristics of the study population at baseline, by treatment.**

| | RSV/TICA | RSV/CLOP | SIMVA/E/TICA | SIMVA/E/CLOP | p-value |
|---|---|---|---|---|---|
| | N = 30 | N = 31 | N = 33 | N = 29 | |
| Age, years | 57 (51–63) | 59 (55–64) | 57 (53–65) | 55 (49–65) | 0.69 |
| Male gender | 22 (73) | 22 (71) | 25 (76) | 23 (79) | 0.99 |
| Diabetes | 11 (37) | 8 (26) | 8 (24) | 9 (31) | 0.81 |
| Smoking | 10 (33) | 12 (39) | 11 (33) | 13 (45) | 0.88 |
| Hypertension | 8 (27) | 8 (26) | 9 (27) | 7 (24) | 0.99 |
| BMI, kg/m$^2$ | 25.9 (23.8–29.4) | 26.6 (24.2–30.2) | 27.1 (23.2–30.9) | 26.7 (24.1–29.9) | 0.88 |
| SBP, mm Hg | 123 (117–131) | 130 (112–145) | 120 (105–133) | 125 (105–144) | 0.64 |
| DBP, mm Hg | 73 (69–82) | 79 (70–87) | 75 (64–81) | 77 (67–89) | 0.96 |
| HbA1c, % | 6.05 (5.57–7.00) | 5.95 (5.53–7.35) | 5.90 (5.50–6.55) | 5.80 (5.35–6.85) | 0.68 |
| Cholesterol | 201 (174–226) | 179 (153–207) | 191 (179–239) | 193 (173–221) | 0.43 |
| LDL-C | 131(109–155) | 121 (102–140) | 130 (104–145) | 121 (105–150) | 0.64 |
| HDL-C | 39 (31–46) | 38 (33–43) | 44 (36–49) | 41 (33–46) | 0.21 |
| Triglycerides | 151 (107–203) | 108 (77–171) | 143 (91–251) | 136 (88–186) | 0.31 |
| Non-HDL-C | 163 (141–187) | 146 (122–166) | 154 (139–195) | 145 (132–181) | 0.45 |

RSV–rosuvastatin; TICA–ticagrelor; SIMVA/E–simvastatin/ezetimibe; CLOP–clopidogrel. BMI–body mass index; SBP–systolic blood pressure; DBP–diastolic blood pressure; HbA1c –glycated hemoglobin. Lipid variables are mg/dL. Values are median (IQR) or n (%). Categorical variables were compared by the Pearson's Chi-square test, and continuous variables by the Kruskal-Wallis test.

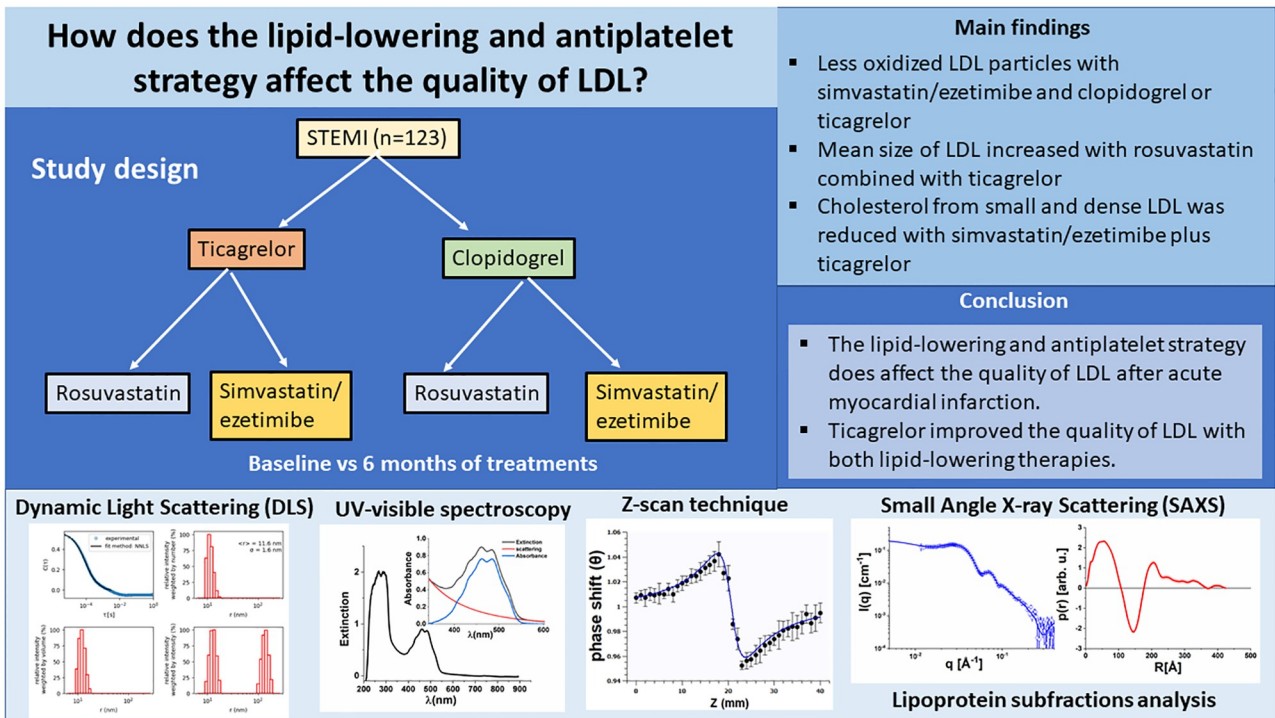

**Fig 2. Graphical abstract.**

Patients receiving clopidogrel increased their θ values from T0 to T6 (0.017 [0.006–0.029] to 0.021 [0.013–0.035], p = 0.024) while no differences were observed for those treated by ticagrelor (0.008 [0.004–0.017] to 0.015 [0.008–0.026], p = 0.060) (Fig 3A).

The results with combined therapies were also examined. Patients treated with rosuvastatin plus clopidogrel did not change their θ values from T0 to T6 (0.019 [0.005–0.030] to 0.020 [0.012–0.034] p = 0.305), as well as those treated with rosuvastatin plus ticagrelor (0.012 [0.004–0.025] to 0.012 [0.006–0.022], p = 1). Conversely, those patients treated with simvastatin/ezetimibe plus clopidogrel had their θ values increased from T0 to T6 (0.016 [0.005–0.023] to 0.023 [0.013–0.035], p = 0.040), as well as those treated with simvastatin/ezetimibe plus ticagrelor (0.006 [0.002–0.013] to 0.015 [0.010–0.029], p = 0.009) (Fig 3B).

## UV-visible spectroscopy

The absorbance values of LDL solution samples at the wavelength corresponding to the maximum of the absorbance spectrum of carotenoids, λ = 484 nm, were examined at T0 and T6. We found no changes in patients treated with rosuvastatin from T0 to T6 (median [IQR]) (0.184 [0.111–0.228] to 0.172 [0.117–0.256], p = 0.817), as well as for those patients treated with simvastatin/ezetimibe (0.147 [0.119–0.250] to 0.196 [0.141–0.252], p = 0.184). Regarding the antiplatelet therapies, those patients treated with clopidogrel did not change their absorbance from T0 to T6 (0.158 [0.111–0.244] to 0.172 [0.126–0.262], p = 0.226), as well as those treated with ticagrelor (0.181 [0.120–0.248] to 0.184 [0.128–0.236], p = 0.955). Likewise, there were no significant differences for combined therapy (all p>0.05).

Correlations between θ and light absorbance at 484 nm.

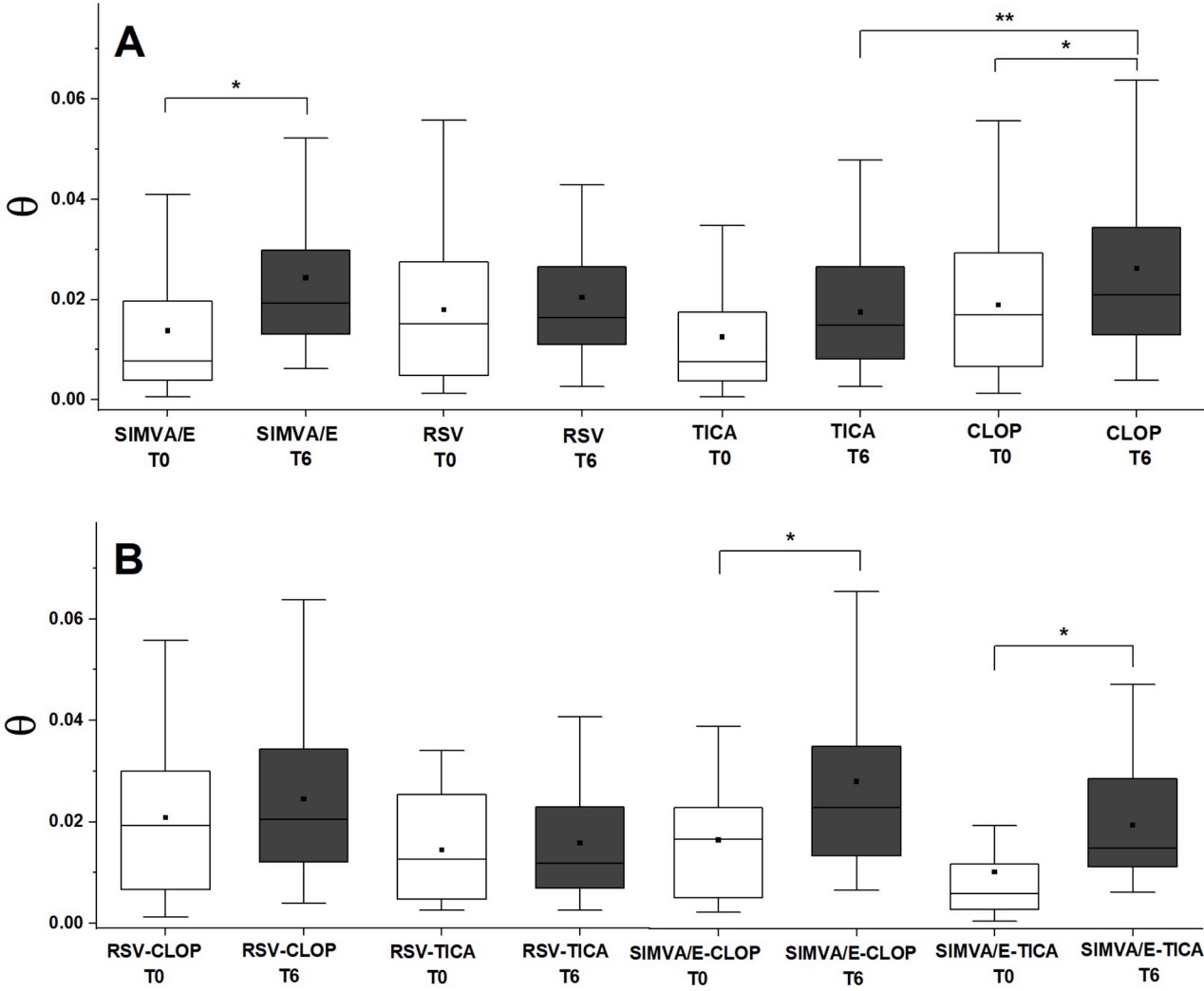

**Fig 3. Box-plot for phase shift (θ) at baseline (T0) and after six months (T6).** (A) according to lipid-lowering and antiplatelet drugs; (B) for combined therapies. Simvastatin/ezetimibe (SIMVA/E), rosuvastatin (RSV), ticagrelor (TICA) and clopidogrel (CLOP). (*: significant difference, Wilcoxon signed rank test, **: significant difference, Mann-Whitney U test).

The θ values (Z-scan) and absorbance (at $\lambda = 484$ nm) were examined for correlations (Spearman's rank test) for each therapy ($\rho_s$).

For simvastatin/ezetimibe, at T0, the correlation between these variables (θ and absorbance 484) was $\rho_s = 0.45$ (p = 0.007), and at T6, $\rho_s = 0.81$ (p = 3.20E-9). For rosuvastatin, the same analysis revealed, at T0, $\rho_s = 0.55$ (p = 1.80E-4), and at T6, $\rho_s = 0.78$ (p = 2.46E-9).

For clopidogrel, at T0, the correlation between these variables was $\rho_s = 0.58$ (p = 7.11E-5), and at T6, $\rho_s = 0.88$ (p = 6.30E-14). Finally, for ticagrelor, at T0, the correlation was $\rho_s = 0.44$ (p = 0.009), and at T6, $\rho_s = 0.72$ (p = 1.26E-6).

Fig 4 shows that for combined therapies at T6, all combined groups presented a significant correlation. The value of θ is strongly correlated to the value of the absorption at wavelength 484 nm that indicates that the higher the θ, the higher the absorption and the higher the number of carotenoids in the LDL particles. This fact implies that the LDL particles are more protected against oxidation as their θ values increase.

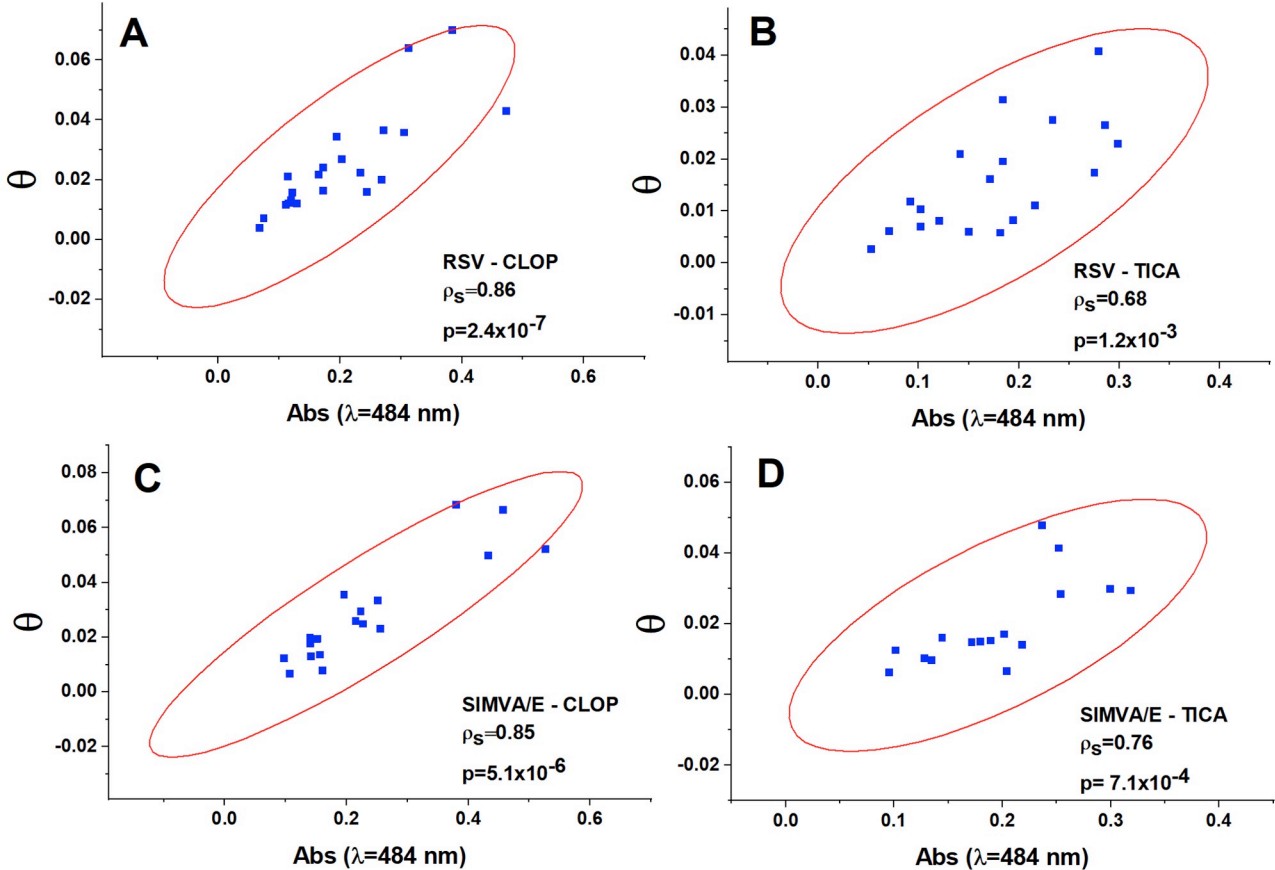

**Fig 4. Correlations between absorbance at wavelength 484 nm and phase shift (θ) after six months of combined therapy (T6).** (A) rosuvastatin-clopidogrel (RSV-CLOP); (B) rosuvastatin-ticagrelor (RSV-TICA); (C) simvastatin/ezetimibe-clopidogrel (SIMVA/E-CLOP); (D) simvastatin/ezetimibe-ticagrelor (SIMVA/E-TICA). Significant correlations were observed for all the combined therapies at T6 (Spearman rank test).

Taking into account the values of θ we can conclude that:

1. Rosuvastatin vs. simvastatin/ezetimibe
   Greater θ at T6 compared to T0 was observed in patients treated with simvastatin/ezetimibe;

2. Clopidogrel vs. ticagrelor
   Greater θ at T6 compared to T0 was observed in patients treated with clopidogrel;
   Simvastatin/ezetimibe- clopidogrel and simvastatin/ezetimibe-ticagrelor
   Greater θ at T6 compared to T0 was observed in patients treated with simvastatin/ezetimibe-clopidogrel and simvastatin/ezetimibe-ticagrelor.

## Small angle X-ray scattering (SAXS)

This technique was used to evaluate lipoproteins in solution, providing information about size, polydispersity, shape, oligomerization, flexibility, contrast of the electronic density of different parts of the particles, and aggregate state. The exposure to lipid-lowering therapies (independent of the antiplatelet therapy) did not change the $R_{LDL}$ (radius of the particle, nm), with both treatments showing similar values (median [IQR]) for this parameter at T0 and T6

with rosuvastatin (13.1 [12.7–13.2] to 13.1[12.8–13.5], p = 0.149) as well as with simvastatin/ezetimibe (13.0 [12.7–13.4] to 13.0 [12.7–13.1], p = 0.192), without differences between groups at T0 or T6. Similar findings were observed for $\mu_{LDL}$ (relative electronic density contrast of the LDL particle), regarding the lipid-lowering therapies. The exposure to rosuvastatin did not change $\mu_{LDL}$ as a function of time (T0 to T6): (-2.42 [-3.00, -2.24] to -2.28 [-2.60, -1.87], p = 0.27). The same result was obtained with the exposure to simvastatin/ezetimibe: (-2.36 [-2.57, -2.16] to -2.56 [-2.88, -2.26], p = 0.06).

Regarding the antiplatelet drugs (independent of lipid-lowering therapy), no differences for $R_{LDL}$ and $\mu_{LDL}$ were found as a function of time, from T0 to T6. For $R_{LDL}$, the exposure to clopidogrel from T0 to T6 (13.1 [12.8–13.4] to 12.9 [12.7–13.2], p = 0.18) was similar to ticagrelor (12.9 [12.6–13.2] to 13.1 [12.9–13.3], p = 0.18). No significant differences in $\mu_{LDL}$ were observed as a function of time for patients exposed to clopidogrel: T6 (-2.37 [-2.52, -2.23]) and T0 (-2.49 [-2.86, -2.26], p = 0.09). The same result was obtained for patients exposed to ticagrelor: T6 (-2.50 [-3.03, -2.23]) and T0 (-2.27 [-2.74, -1.87], p = 0.20).

For the combined therapies, patients receiving rosuvastatin plus ticagrelor significantly increase their values for $R_{LDL}$ (p = 0.03) and decrease their absolute value of the relative electronic density contrast for $|\mu_{LDL}|$ (p = 0.04), while no differences were observed for other groups (rosuvastatin plus clopidogrel, simvastatin/ezetimibe plus ticagrelor or simvastatin/ezetimibe plus clopidogrel). Fig 5 shows $R_{LDL}$ and $|\mu_{LDL}|$ with the different therapies.

Considering all samples together, a correlation between $R_{LDL}$ and $\mu_{LDL}$ was observed at T6 ($\rho_s$ = 0.44, p = 0.039). Furthermore, a correlation between $R_{LDL}$ and DLS results was found at T6 ($\rho_s$ = 0.44, p = 0.04).

These results suggest that the dimensions of LDL, as well as its electron density distribution, do not change as a result of antiplatelet or statin-based therapies, when the effects of drugs are analyzed individually. For the combination of drugs, i.e., statin-antiplatelet therapies, there were significant differences and the results indicated that:

$R_{LDL}$: rosuvastatin-ticagrelor > rosuvastatin-clopidogrel;

$R_{LDL}$: rosuvastatin-ticagrelor > simvastatin/ezetimibe-ticagrelor;

$R_{LDL}$: T6—rosuvastatin-ticagrelor > T0—rosuvastatin-ticagrelor;

$\mu_{LDL}$: rosuvastatin-ticagrelor > rosuvastatin-clopidogrel;

$\mu_{LDL}$: rosuvastatin-ticagrelor > simvastatin/ezetimibe-ticagrelor;

$\mu_{LDL}$: T6—rosuvastatin-ticagrelor > T0 –rosuvastatin-ticagrelor.

Let us assume that the higher the LDL mean radius measured by SAXS, the less atherogenic the LDL is (since small and dense particles are the most atherogenic). With respect to this parameter, the treatment rosuvastatin-ticagrelor was shown to be the most efficient, since the mean size of the LDLs increased after six months of the treatment. Moreover, considering that in the early stages of LDL oxidation (represented in the following by the super index $^*$), there are no important modifications in the core of the particle, we expect that $\Delta\rho^*_{core} \sim \Delta\rho_{core} \sim constant$ but, in the shell, modifications on the electric charger of the particles occur (since more atherogenic LDLs are electronegative), and $\Delta\rho^*_{shell} > \Delta\rho_{shell}$ [28]. In this framework, we expect that the absolute value of $\mu_{LDL}$ increases after the treatment (T6), as an efficient response to this parameter.

Another finding is the occurrence of a positive correlation between the SAXS parameters, $R_{LDL}$ and $\mu_{LDL}$, based on statistical tests. Considering all samples together, a correlation between $R_{LDL}$ and $\mu_{LDL}$ was observed at T6 ($\rho_s$ = 0.44, p = 0.039).

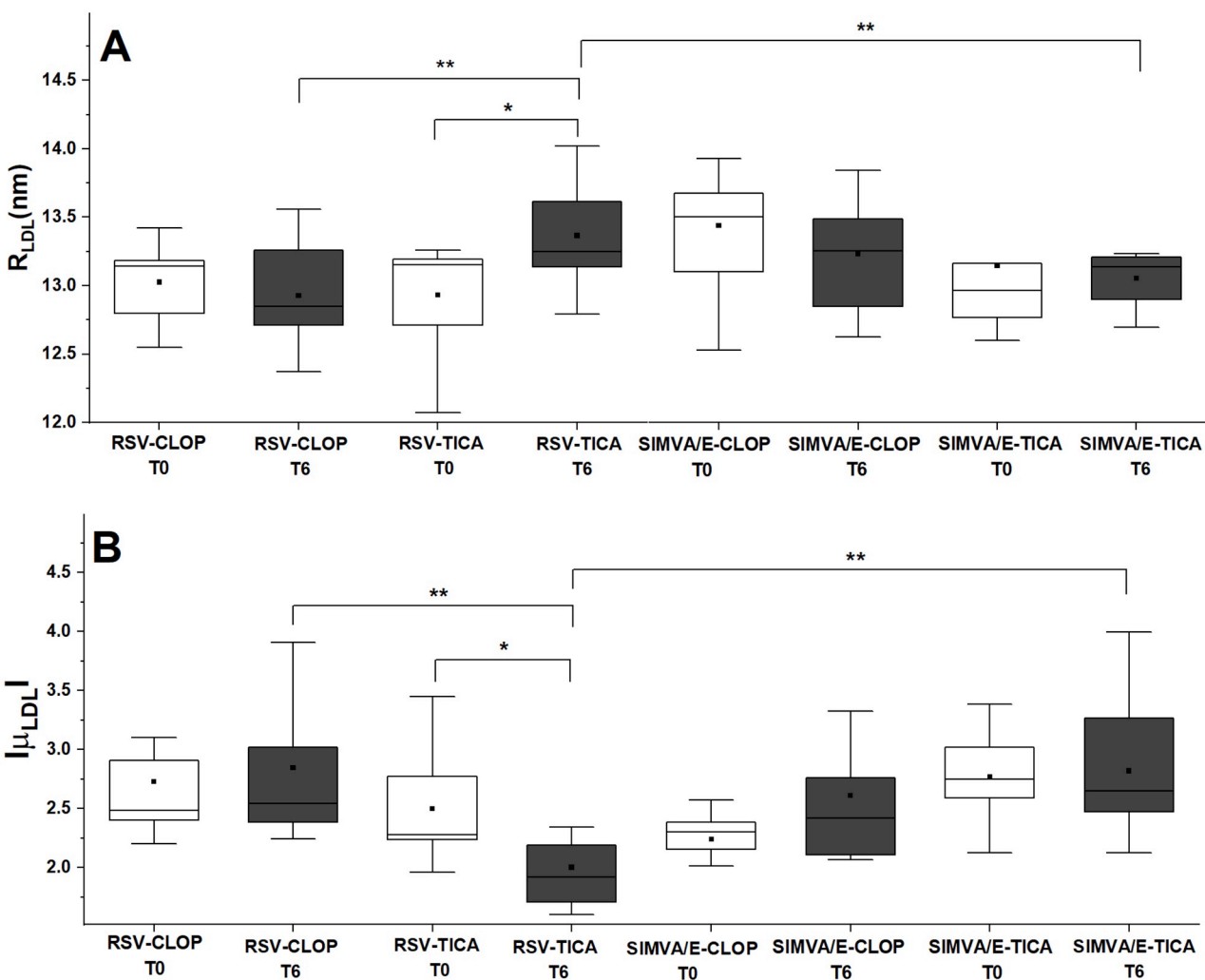

**Fig 5. Box-plots for combined therapies at baseline (T0) and after six months (T6).** (A) radius of the LDL particles $R_{LDL}$; (B) relative electronic density contrast of the LDL particles $|\mu_{LDL}|$. Simvastatin/ezetimibe (SIMVA/E), rosuvastatin (RSV), ticagrelor (TICA) and clopidogrel (CLOP). (*: significant difference, Wilcoxon signed rank test and **: significant difference, Kruskal-Wallis test with Dunn's post-hoc test, $p<0.05$).

### Dynamic light scattering (DLS)

The DLS was used to assess the eventual aggregation of particles and the LDL size distribution. There was a significant increase in the R (radius of the particle in nm) for those patients receiving rosuvastatin, from T0 to T6 (median [IQR]), (10.9 [9.9–12.3] to 11.2 [10.8–12.3], $p = 0.026$), while no differences in DLS were observed for those patients treated with simvastatin/ezetimibe (10.6 [9.8–11.3] to 10.8 [10.2–11.3], $p = 0.407$). No differences between lipid-lowering groups were observed at T0, but higher values for particle size from DLS were observed at T6 for patients treated with rosuvastatin ($p = 0.006$).

The same evaluation was made according to the antiplatelet therapies. A significant increase in the R (nm) was observed for those treated with clopidogrel (10.2 [9.8–11.4] to 11.1 [10.7–11.7], $p = 0.034$). Contrarily, patients treated by ticagrelor did not change their particle size by DLS (11.1 [10.2–11.9] to 11.0 [10.3–12.1], $p = 0.449$). However, no significant differences between antiplatelet groups were observed at T0 or T6.

Regarding the combined treatments, the mean value of the LDL particle size at T6 was higher in the rosuvastatin-ticagrelor group (11.8 [10.9–12.7]) than simvastatin/ezetimibe-ticagrelor (10.2 [10.0–10.6], p = 0.04), and simvastatin/ezetimibe-clopidogrel (11.2 [10.7–11.7], p<0.001) groups.

With respect to the DLS results, our findings indicate that:

1. Rosuvastatin increases the particle size from T0 to T6, which should be interpreted as a favourable response;

2. Rosuvastatin was more effective than simvastatin/ezetimibe, exhibiting a larger particle diameter at T6;

3. Clopidogrel was also more effective than ticagrelor from T0 to T6, being associated with a larger size of LDLs;

4. Mean value of the LDL particle size at T6 was significantly higher in rosuvastatin-ticagrelor group than simvastatin/ezetimibe-ticagrelor or simvastatin/ezetimibe-clopidogrel groups.

### Classic lipid profile and subfractions of LDL

Table 1 shows the classic lipid profile of the population at baseline, by treatment. No differences for total cholesterol, LDL-C, HDL-C, triglycerides, or for non-HDL-C were observed at T0. At T6 all combined treatments were associated with significant changes in the lipid profile for total cholesterol, LDL-C and non-HDL-C, but no differences were seen for HDL-C. Interestingly, a decrease in triglycerides was observed only for combined therapies with ticagrelor (rosuvastatin-ticagrelor and simvastatin/ezetimibe-ticagrelor) (Fig 6).

The subfractions analysis showed that patients treated with simvastatin/ezetimibe had a decrease in the cholesterol content of large LDL particles (median [IQR]) from T0 to T6 (47.2 [34.1–61.0] to 28.4 [22.5–35.7], p<0.001), but not for the small dense LDL particles (2.9 [0.0–8.3] to 2.3 [0.0–4.2], p = 0.062). For those patients treated with rosuvastatin, there was a significant decrease in the cholesterol content of the large LDL particles from T0 to T6 (54.0 [37.4–64.3] to 22.9 [21.0–33.4], p<0.001), but not for small dense LDL particles (2.9 [0.0–5.0] to 1.4 [0.0–5,7], p = 0.793). The same analysis for the antiplatelet therapies showed, for those patients treated with clopidogrel, a decrease in the cholesterol content of the large LDL particles (48.3 [34.0–61.2] to 27.0 [20.5–33.7], p<0.001), but no changes in the cholesterol content of small and dense LDL particles at T0 compared to T6 (2.5 [0.0–5.1] to 2.2 [0.0–7.8], p = 0.769). For those patients treated with ticagrelor there was a significant decrease in the large LDL particles from T0 to T6 (52.6 [41.2–63.7] to 24.7 [21.8–35.1], p<0.001), as well as for the small dense LDL particles (3.1 [0.0–7.7] to 1.6 [0.0–3.8], p = 0.021).

Considering the percentage of the large LDL particles, those patients treated with simvastatin did not change the distribution of these particles from T0 to T6 (p = 0.15), while there was a significant decrease in the percentage of these particles with rosuvastatin after six months (p = 0.004). The same analysis for the antiplatelet therapies showed no differences for those treated with clopidogrel, while a decrease in the percentage of less atherogenic LDL particles was observed in those patients treated with ticagrelor (p = 0.001).

Significant correlations between the percentage and cholesterol content of LDL subfractions and $R_{LDL}$ were obtained at T0 and T6. At T0, a negative correlation between $R_{LDL}$ and cholesterol content of lipoprotein was found for LDL3 ($\rho_s$ = -0.51, p = 0.013), and LDL4 ($\rho_s$ = -0.58, p = 0.004). Considering the cholesterol content of small dense LDL subfractions together (LDL3 to LDL7), the correlation obtained with $R_{LDL}$ at T0 was ($\rho_s$ = -0.58, p = 0.006). No significant correlations were seen at T6.

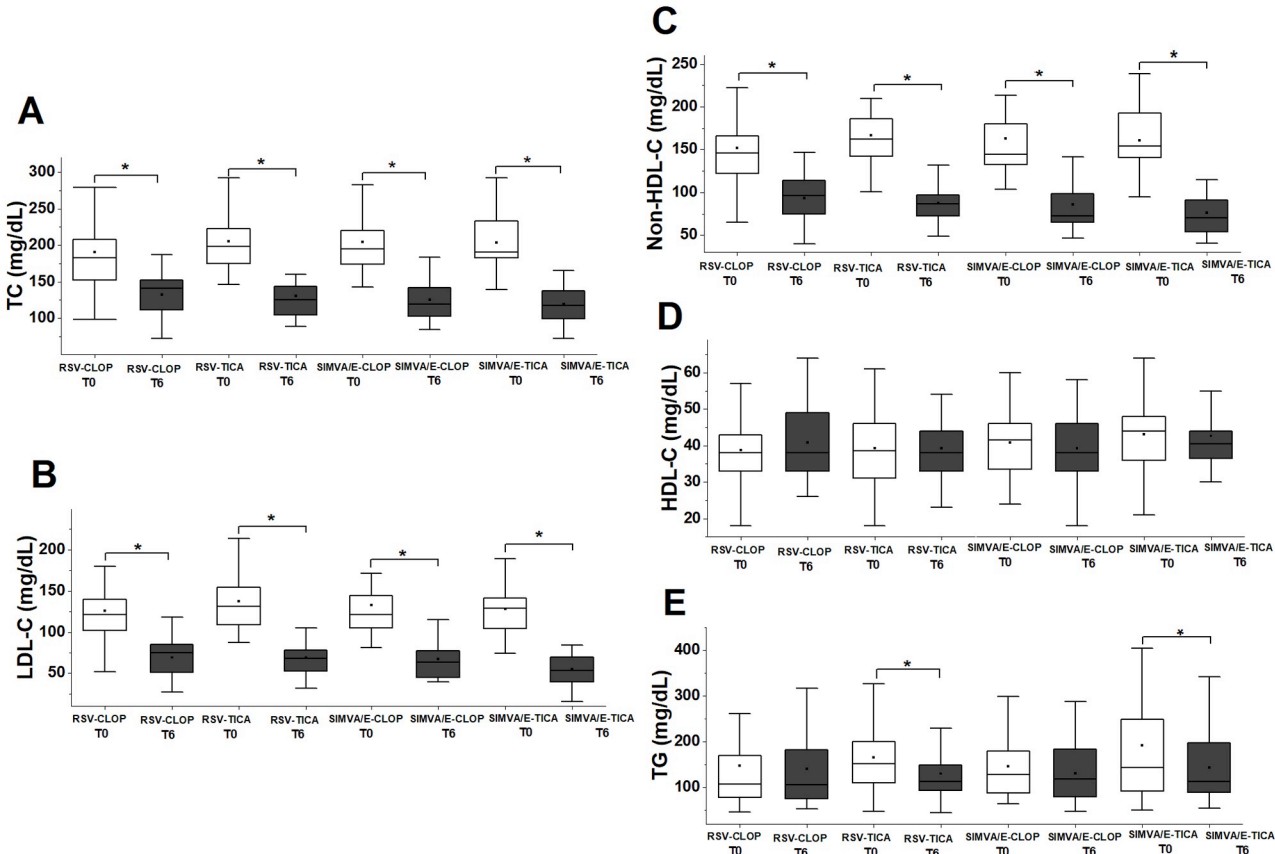

**Fig 6. Classic lipid profile at baseline (T0) and after six months (T6).** (A) Total cholesterol at T0 and at T6 with combined therapy. Significant decrease was observed with all the combined therapies. For rosuvastatin-clopidogrel (p = 7.4E-9); for rosuvastatin-ticagrelor (p = 3.7E-9); for simvastatin/ezetimibe-clopidogrel (p = 1.4E-8); and for simvastatin/ezetimibe-ticagrelor (p = 1.8E-9). (B) LDL-C at T0 and at T6 with combined therapy. Significant decrease was observed with all the combined therapies. For rosuvastatin-clopidogrel (p = 1.8E-9); for rosuvastatin-ticagrelor (p = 3.7E-9); for simvastatin/ezetimibe-clopidogrel (p = 5.9E-8); and for simvastatin/ezetimibe-ticagrelor (p = 4.6E-10). (C) Non-HDL-C at T0 and at T6 with combined therapy. Significant decrease was observed with all the combined therapies. For rosuvastatin-clopidogrel (p = 2.9E-8); for rosuvastatin-ticagrelor (p = 3.4E-9); for simvastatin/ezetimibe-clopidogrel (p = 2.9E-8); and for simvastatin/ezetimibe-ticagrelor (p = 4.6E-10). (D) HDL-C at T0 and at T6 with combined therapy. None of the combined therapies significantly changed HDL-C. For rosuvastatin-clopidogrel (p = 0.1580); for rosuvastatin-ticagrelor (p = 0.9062); for simvastatin/ezetimibe-clopidogrel (p = 0.6402); and for simvastatin/ezetimibe-ticagrelor (p = 0.1750). (E) Triglycerides at T0 and at T6. Only the combined groups with ticagrelor showed significant decrease in triglycerides. For rosuvastatin-clopidogrel (p = 0.667); for rosuvastatin-ticagrelor (p = 0.020); for simvastatin/ezetimibe-clopidogrel (p = 0.640); and for simvastatin/ezetimibe-ticagrelor (p = 0.006). RSV–rosuvastatin; SIMVA/E–simvastatin/ezetimibe; CLOP–clopidogrel; TICA–ticagrelor. (*: significant difference, Wilcoxon signed rank test).

With regard to combined therapies, a decrease in the cholesterol from large LDL particles was seen from T0 to T6 in the rosuvastatin/ticagrelor group (55.6 [44.6–69.1] to 23.0 [21.8–36.8], p<0.001), but not for small and dense LDL particles (3.0 [0.0–7.3] to 1.3 [0.0–3.8], p = 0.173) (Fig 7A). It was observed a decrease in the large LDL for the rosuvastatin/clopidogrel group (52.1 [35.8–62.4] to 22.7 [19.3–30.9], p<0.001), while no significant changes were seen for small and dense LDL particles (0.0 [0.0–4.0] to 2.2 [0.0–10.4], p = 0.407) (Fig 7B). However, for the simvastatin/ezetimibe/ticagrelor group, a significant decrease in the cholesterol content of large LDL was seen (51.1 [39.9–61.4] to 25.7 [21.4–33.7], p = 0.001, as well as for the small and dense LDL particles (4.8 [0.0–7.9] to 2.3 [0.0–4.2], p = 0.044) (Fig 7C). Finally, for the simvastatin/ezetimibe/clopidogrel group, there was a decrease in the large LDL (42.2 [32.7–59.5] to 29.0 [23.5–38.6], p = 0.028), but not for the small and dense LDL (2.6 [0.0–

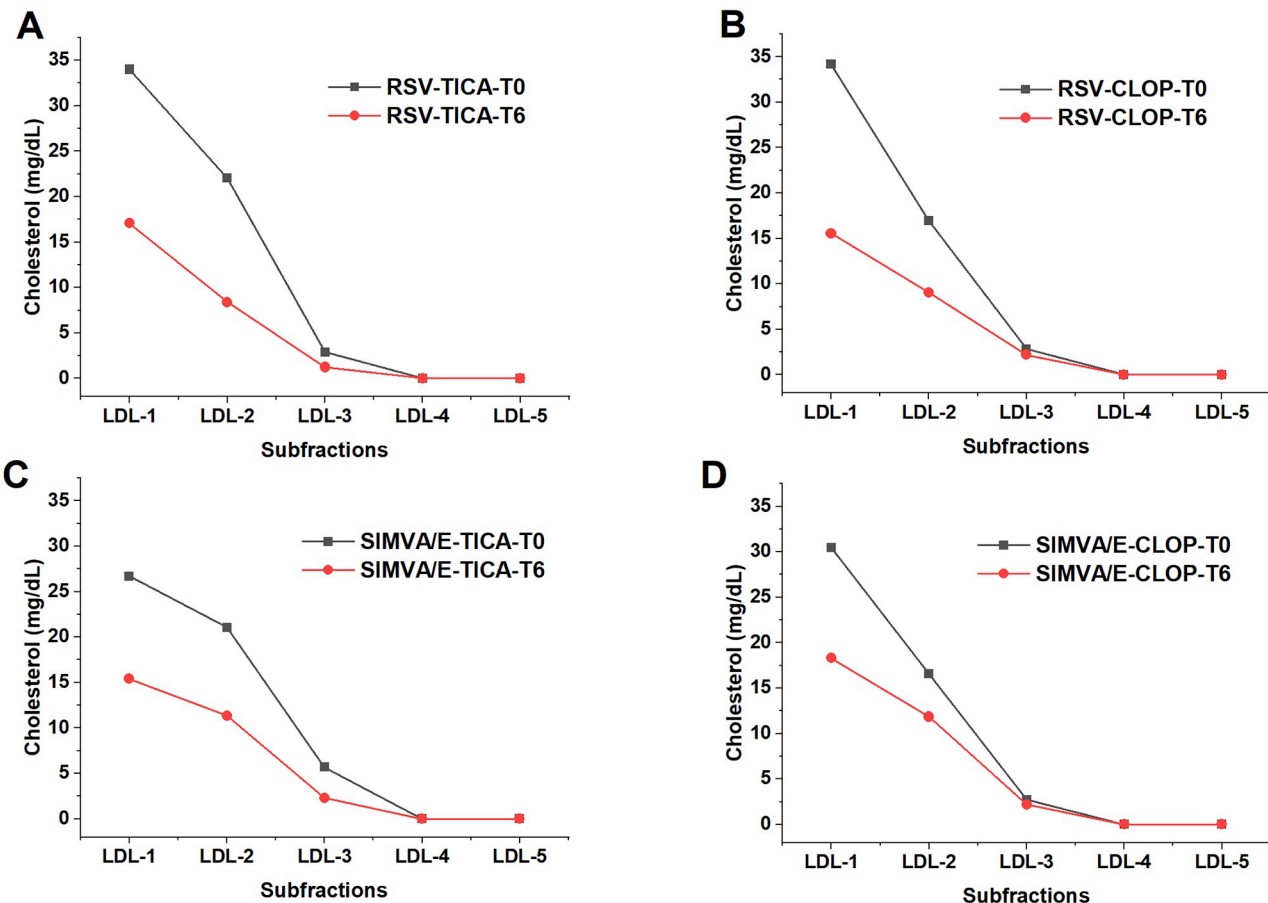

**Fig 7. LDL subfractions at baseline (T0) and after six months (T6).** (A) Group rosuvastatin-ticagrelor (RSV/TICA). The combined therapy decreased the cholesterol content of large and buoyant LDL particles (LDL1 and LDL 2) (p<0.001), but not from the small and dense LDL particles (LDL3-LDL7). (B) Group rosuvastatin-clopidogrel (RSV/CLOP). The combined therapy decreased the cholesterol content of large and buoyant LDL particles (LDL1 and LDL 2,) (p<0.001) but not from the small and dense LDL particles (LDL3-LDL7). (C) Group simvastatin/ezetimibe-ticagrelor (SIMVA/E-TICA). The combined therapy decreased the cholesterol content of large and buoyant LDL particles (LDL1 and LDL 2) (p = 0.001), and also from the small and dense LDL particles (LDL3-LDL7) (p = 0.044). (D) Group simvastatin/ezetimibe-clopidogrel (SIMVA/E-CLOP). The combined therapy decreased the cholesterol content of large and buoyant LDL particles (LDL1 and LDL 2) (p = 0.028), but not from the small and dense LDL particles (LDL3-LDL7).

8.8] to 2.2 [0.4–6.2], p = 0.654) (Fig 7D). Considering the percentage of LDL particles, the only significant change was in the group rosuvastatin/ticagrelor, where a percent decrease in the large and buoyant LDL (p = 0.02) was seen.

Conclusions of LDL subfractions data for combined therapies:

1. Simvastatin/ezetimibe-ticagrelor therapy was associated with a decrease in the cholesterol content of the large LDL and from the small and dense LDL subfractions;

2. Simvastatin/ezetimibe-clopidogrel therapy was associated with a decrease in the cholesterol content only for the large LDL;

3. Rosuvastatin-ticagrelor therapy was associated with a decrease in the cholesterol content only for large LDL;

4. Rosuvastatin-clopidogrel was associated with a decrease in the cholesterol content only for the large LDL.

## Discussion

According to current guidelines [1, 2], all patients with STEMI should be treated with lipid-lowering and antiplatelet drugs. Low cholesterol levels and low platelet activation are recommended, but the effects of these therapies on the quality of lipoproteins are much less reported, especially when combined. This study examined the effects of drugs commonly used after acute coronary syndrome: rosuvastatin, simvastatin plus ezetimibe, clopidogrel, and ticagrelor. After six months of randomized exposure to these antiplatelet and lipid-lowering therapies, many differences in the quality of lipoproteins were observed with various techniques, such as assessment of nonlinear optical properties of LDL (Z-scan), small angle X-ray scattering (SAXS), dynamic light scattering (DLS), UV-visible spectroscopy, and by the LDL subfractions electrophoresis analysis. There were correlations between characteristics of LDL particle seen by these methods and also differences according to each therapy, alone or combined. These analyses provide information regarding the quality of LDL, oxidation, pattern of aggregation, particle size and distribution into subfractions (large and buoyant or small and dense). The period of six months was chosen to analyze the quality of LDL after the acute inflammatory phase of myocardial infarction.

One important finding of our study was the greater effectiveness of the simvastatin/ezetimibe plus either clopidogrel or ticagrelor with respect to the best quality and functionality of LDL after six months with these treatments. This result can be interpreted based on the number of carotenoids present in the LDL particles. A possible explanation for this finding is the higher lipophilicity and antioxidant effect on lipoproteins reported for simvastatin [29]. While rosuvastatin reduces LDL-C due to higher LDL receptor expression, the simvastatin/ezetimibe combination has other mechanisms such as lower intestinal cholesterol absorption. Thus, due to the greater affinity of LDL receptors for larger and buoyant LDLs, the effects of rosuvastatin are more pronounced in the removal of less atherogenic LDL particles. Interestingly, activated platelets markedly increase internalization of oxidized LDL [30]. Therefore, antiplatelet therapy may influence the pattern of LDL subfractions not only due to direct antioxidant or anti-inflammatory properties but also through the removal of oxidized LDL by activated platelets. Thus, the carotenoid content can be spared in LDL particles through these mechanisms, explaining our Z-scan results. The value of θ was strongly correlated to the value of the absorption at 484 nm indicating that the higher the θ, the higher the absorption and the higher the number of carotenoids in the LDL particles. This fact implies that the LDL particles are more protected against oxidation the higher their θ value measured.

In summary, in terms of the optical parameters, which reflect the functionality of the LDL (more protected against oxidation, i.e., more carotenoids present in the LDL structure), the combination simvastatin/ezetimibe plus clopidogrel or ticagrelor seems to be more efficient with respect to the LDL functionality. However, exposure to a combined therapy of rosuvastatin with ticagrelor was shown to increase more efficiently the size of LDL particles, based on the SAXS and DLS parameters, a characteristic of less atherogenic particle.

In addition to some differences in the antiplatelet activity, clopidogrel and ticagrelor may differ in their antiatherogenic effects. Interestingly, the P2Y[12] receptors are not expressed exclusively in platelets, but also in endothelial cells, particularly in culprit coronary plaques [31]. Treatment with ticagrelor in subjects with acute coronary syndromes was associated with improvement in the endothelial function and decrease in inflammatory markers [32]. Furthermore, ticagrelor decreases PCSK9 expression, a key protein in the LDL receptor catabolism.[7] Thus, the use of ticagrelor may be associated with more efficient clearance of LDL particles, mainly those large and buoyant, which has been reported with rosuvastatin and PCSK9 inhibitor [10, 11, 33]. In addition, ticagrelor has been associated with a possible anti-atherosclerotic

effect via the higher activity of the antioxidant enzyme paraoxonase 1 (PON-1) [34]. Interactions between clopidogrel and statins, particularly for those sharing the same pathway for metabolization, raised concerns for pharmacokinetic interactions. However, the use of statins, with or without metabolism via CYP3A4, has been reported as safe, not influencing platelet activation or cardiovascular events [35, 36]. Nevertheless, increased plasma concentrations of rosuvastatin were reported after concomitant exposure to clopidogrel [37]. Ticagrelor may influence the removal of large and buoyant LDL particles through PCSK9 inhibition, favouring the recycling of LDL receptors [7, 38]. In fact, based on the LDL subfractions analysis, the more efficient percent decrease in the large and buoyant LDL was observed with the combined therapy with rosuvastatin and ticagrelor.

## Limitations

Our study included a relatively small sample of patients, but the data was obtained by a variety of techniques showing uniform results. The trial was open label, but it was randomized and the LDL quality was analyzed blindly. In addition, the study compared individuals with very similar baseline characteristics. These characteristics are routinely found in the general population with their first myocardial infarction.

## Conclusions

After acute myocardial infarction, the effects on the quality of LDL particles were influenced by the antiplatelet/lipid-lowering strategy. The combination of simvastatin/ezetimibe with either clopidogrel or ticagrelor was associated with less oxidized LDL particles, and when simvastatin/ezetimibe was combined with ticagrelor, it lowered the cholesterol amount distributed in atherogenic subfractions of LDL. The combination of rosuvastatin with ticagrelor was associated with increase in the size of LDL particles, also a favorable response. Thus, considering these analyses together, the LDL quality was improved with the use of ticagrelor with both lipid-lowering therapies. These interesting findings and their relevance should be examined in larger cardiovascular outcome trials.

## Supporting information

**S1 Checklist.**
(DOC)

**S1 Data.**
(XLSX)

**S1 File.**
(DOCX)

**S2 File.**
(DOCX)

## Author Contributions

**Conceptualization:** Francisco A. H. Fonseca, Antônio M. F. Neto.

**Data curation:** Zahra Lotfollahi, Francisco A. H. Fonseca, Antônio M. F. Neto.

**Formal analysis:** Zahra Lotfollahi, Ana P. Q. Mello, Francisco A. H. Fonseca, Luciene O. Machado, Andressa F. Mathias, Maria C. Izar, Nagila R. T. Damasceno, Cristiano L. P. Oliveira, Antônio M. F. Neto.

**Funding acquisition:** Francisco A. H. Fonseca, Antônio M. F. Neto.

**Investigation:** Zahra Lotfollahi, Ana P. Q. Mello, Francisco A. H. Fonseca, Luciene O. Machado, Andressa F. Mathias, Maria C. Izar, Nagila R. T. Damasceno.

**Methodology:** Zahra Lotfollahi, Ana P. Q. Mello, Luciene O. Machado, Maria C. Izar, Nagila R. T. Damasceno, Cristiano L. P. Oliveira, Antônio M. F. Neto.

**Project administration:** Francisco A. H. Fonseca, Antônio M. F. Neto.

**Resources:** Francisco A. H. Fonseca.

**Software:** Zahra Lotfollahi, Nagila R. T. Damasceno, Cristiano L. P. Oliveira, Antônio M. F. Neto.

**Supervision:** Francisco A. H. Fonseca, Maria C. Izar, Nagila R. T. Damasceno, Cristiano L. P. Oliveira, Antônio M. F. Neto.

**Validation:** Ana P. Q. Mello, Maria C. Izar, Nagila R. T. Damasceno, Cristiano L. P. Oliveira, Antônio M. F. Neto.

**Writing – original draft:** Francisco A. H. Fonseca.

**Writing – review & editing:** Zahra Lotfollahi, Ana P. Q. Mello, Francisco A. H. Fonseca, Luciene O. Machado, Maria C. Izar, Nagila R. T. Damasceno, Antônio M. F. Neto.

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
