## [Decision Letter · Decision Letter 0]

5 Apr 2022

PONE-D-21-17809Changes in lipoproteins associated with pharmacological strategies in patients with acute myocardial infarctionPLOS ONE

Dear Dr. Fonseca,

Thank you for submitting your manuscript to PLOS ONE. After careful consideration, we feel that it has merit but does not fully meet PLOS ONE’s publication criteria as it currently stands. Therefore, we invite you to submit a revised version of the manuscript that addresses the points raised during the review process.

Please read carefully all comments bt the external expert reviewers and address them one by one.

We look forward to receiving your revised manuscript.

Kind regards,

Salvatore De Rosa

Academic Editor

PLOS ONE

Journal Requirements:

2. Thank you for stating the following in the Competing Interests section: "FF has served as steering committee member of the JUPITER trial (AstraZeneca) and CANTOS trial (Novartis). FF reports receiving fees as speaker for Sanofi, AstraZeneca, Amgen, Novo Nordisk, Libbs."

We note that you received funding from a commercial source: AstraZeneca, Novartis, Sanofi, Amgen, Novo Nordisk, Libbs

4. Please upload a new copy of Figures 1 and 2 as the detail is not clear. Please follow the link for more information: https://blogs.plos.org/plos/2019/06/looking-good-tips-for-creating-your-plos-figures-graphics/" https://blogs.plos.org/plos/2019/06/looking-good-tips-for-creating-your-plos-figures-graphics/

Reviewers' comments:

Reviewer's Responses to Questions

**Comments to the Author**

1. Is the manuscript technically sound, and do the data support the conclusions?

Reviewer #1: Partly

Reviewer #2: Yes

Reviewer #3: Yes

2. Has the statistical analysis been performed appropriately and rigorously? 

Reviewer #1: No

Reviewer #2: Yes

Reviewer #3: Yes

3. Have the authors made all data underlying the findings in their manuscript fully available?

Reviewer #1: Yes

Reviewer #2: No

Reviewer #3: Yes

4. Is the manuscript presented in an intelligible fashion and written in standard English?

Reviewer #1: Yes

Reviewer #2: Yes

Reviewer #3: Yes

5. Review Comments to the Author

Reviewer #1: PONE-D-21-17809: statistical review

SUMMARY. This is a prospective, open label study of the effects of combined therapies on the quality of low-density lipoprotein (LDL) particles in subjects with acute myocardial infarction. Groups seem well randomized. Correctly, cross-sectional comparisons are made by unpaired t-tests or Mann-Whitney U tests, while longitudinal comparisons are made by paired t-tests or Wilcoxon tests. I however list below some specific points that should be addressed.

SPECIFIC POINTS

1. Follow-up duration (6 months) is not motivated; could results change under different durations? Can the authors motivate such follow-up duration?

2. This is an open label study. As such, it is liable to ascertainment bias. This limitation should be remarked in the final discussion.

3. In this paper, all the statistical tests are two-tailed (as stated in the statistical analysis section). However, one-tail tests seem more appropriate here as the alternative hypothesis is often one-directional. For example, on page 10, the authors say that they observed a significant “increase” of theta. In this case, the alternative hypothesis is not “theta at T0 is different of theta at T6” but rather “theta at T6 is greater than theta at T0”. I would suggest to use a two-tailed test when the alternative hypothesis is that the two parameters are different and a one-tailed test when the alternative hypothesis is that a parameter is greater than another.

Reviewer #2: Research article titled (Changes in lipoproteins associated with pharmacological strategies in patients with

acute myocardial infarction) is an interesting one with high clinical potential. However, the presentation of the article needs revision starting from the title.

1- Title: authors better should identify what medications followed up in their study.

2-Clear exclusion and exclusion criteria should be stated

3- Stat analysis: authors mentioned tests were applied when appropriate: please give a clear statement for application of the 2 tests.

4- Most of abbreviations are not explained at the first appearance.

5- Use appropriate abbreviations (e.g. h for hours...)

Reviewer #3: Congratulations to all authors involved in this important manuscript entitled "Change in lipoproteins associated with pharmacological strategies in acute myocardial infarction”

Just to make it more understandable to readers, I suggest minor changes to the text.

1- I suggest standardizing the term (STEMI) to “ST-segment elevation myocardial infarction”, Instead of "ST-segment elevation acute myocardial infarction) - line 28

2- Patients who were unsuccessful after thrombolysis and were referred for rescue angioplasty were included in this cohort?

3- Regarding the comparisons between the groups, using the non-parametric test, I suggest some considerations for a better understanding of the data

The Kruskal–Wallis test by ranks, Kruskal–Wallis H test is a non-parametric method for testing whether samples originate from the same distribution. It is used for comparing two or more independent samples of equal or different sample sizes. It extends the Mann–Whitney U test, which is used for comparing only two groups. The parametric equivalent of the Kruskal–Wallis test is the one-way analysis of variance (ANOVA). The test does not identify where this stochastic dominance occurs or for how many pairs of groups stochastic dominance obtains. For analyzing the specific sample pairs for stochastic dominance, Dunn's test, pairwise Mann–Whitney tests with Bonferroni correction.

Thus, in this manuscript, I did not observe the new alpha value, post Bonferroni correction (original alpha divided by the number of comparisons). In this case 0.05/4, which will be the post-test alpha, to accept or reject the null hypothesis (equality).

6. PLOS authors have the option to publish the peer review history of their article (what does this mean?). If published, this will include your full peer review and any attached files.

Reviewer #1: No

Reviewer #2: **Yes: **Sawsan Zaitone

Reviewer #3: **Yes: **Henrique Tria Bianco

---

## [Author Response · Author response to Decision Letter 0]

25 Apr 2022

We revised our manuscript according to the suggestions made by the journal and reviewers. 

Response to reviewers

Reviewer #1: PONE-D-21-17809: statistical review

SUMMARY. This is a prospective, open label study of the effects of combined therapies on the quality of low-density lipoprotein (LDL) particles in subjects with acute myocardial infarction. Groups seem well randomized. Correctly, cross-sectional comparisons are made by unpaired t-tests or Mann-Whitney U tests, while longitudinal comparisons are made by paired t-tests or Wilcoxon tests. I however list below some specific points that should be addressed.

SPECIFIC POINTS

1. Follow-up duration (6 months) is not motivated; could results change under different durations? Can the authors motivate such follow-up duration?

Authors: The period of six months was chosen to analyze the quality of LDL after the acute inflammatory phase. The purpose was to analyze the effects of antiplatelet and lipid-lowering therapies in the longer term to reduce possible effects resulting from the greater inflammatory activity and tissue repair in the early phase of STEMI.

We added in the text (see page 16, lines 29, 30)

The period of six months was chosen to analyze the quality of LDL after the acute inflammatory phase of myocardial infarction.

2. This is an open label study. As such, it is liable to ascertainment bias. This limitation should be remarked in the final discussion.

Authors: We added the information in the study limitations (see page 18, lines 9-11)

The trial was open label, but it was randomized and the LDL quality was analyzed blindly. In addition, the study compared individuals with very similar baseline characteristics.

3. In this paper, all the statistical tests are two-tailed (as stated in the statistical analysis section). However, one-tail tests seem more appropriate here as the alternative hypothesis is often one-directional. For example, on page 10, the authors say that they observed a significant “increase” of theta. In this case, the alternative hypothesis is not “theta at T0 is different of theta at T6” but rather “theta at T6 is greater than theta at T0”. I would suggest to use a two-tailed test when the alternative hypothesis is that the two parameters are different and a one-tailed test when the alternative hypothesis is that a parameter is greater than another.

Authors: Thank you for your important comment. In the captions of the figures, we have indicated which tests were performed. For comparison between theta T6 and T0 we did not use unpaired two sample t-test because theta did not have normal distribution. In this case we used a non-parametric Mann-Whitney U test. (see page 9, caption Fig 3. lines 21-22). We follow your suggestion and changed the text. (see page 10, lines 26-31).

Taking into account the values of θ we can conclude that:

1) Rosuvastatin vs. simvastatin/ezetimibe

 Greater θ at T6 compared to T0 was observed in patients treated with simvastatin/ezetimibe;

2) Clopidogrel vs. ticagrelor

 Greater θ at T6 compared to T0 was observed in patients treated with clopidogrel;

3) Simvastatin/ezetimibe- clopidogrel and simvastatin/ezetimibe-ticagrelor

 Greater θ at T6 compared to T0 was observed in patients treated with simvastatin/ezetimibe-clopidogrel and simvastatin/ezetimibe-ticagrelor.

Reviewer #2: Research article titled (Changes in lipoproteins associated with pharmacological strategies in patients with

acute myocardial infarction) is an interesting one with high clinical potential. However, the presentation of the article needs revision starting from the title.

1- Title: authors better should identify what medications followed up in their study.

Authors: we changed the title according to your suggestion (see page 1, line 1)

Changes in lipoproteins associated with lipid-lowering and antiplatelet strategies in patients with acute myocardial infarction

2-Clear exclusion and exclusion criteria should be stated

Authors: we added information for the inclusion and exclusion criteria to those already described (see page 3, lines 8-13)

All included patients were submitted to pharmacological thrombolysis in the first 6 hours of STEMI and referred to Hospital Sao Paulo to perform coronary angiogram and percutaneous coronary intervention (PCI) when needed, in the first 24 h of STEMI (pharmacoinvasive strategy). Key exclusion criteria were clinical instability, use of lipid-lowering or immunosuppressant therapies, autoimmune disease, known malignancy, pregnancy, or signs of active infections.

3- Stat analysis: authors mentioned tests were applied when appropriate: please give a clear statement for application of the 2 tests.

Authors: we state when the test was applied. (see page 7, lines 4-6) 

For comparisons between groups, unpaired two sample t-test (2-tailed) or the Mann-Whitney U test, were used for variables with normal or non-Gaussian distribution, respectively.

In addition, we have indicated in each figure's caption which tests have been conducted.

4- Most of abbreviations are not explained at the first appearance.

Authors: we explained the abbreviations. (in the Abstract: see page 1, lines 24 and 32, in the Introduction: pages 2, lines 18-22 and 31,32, page 4, in the Materials and methods, Blood samples, lines 21,22) 

low-density lipoprotein (LDL) particles.

ultra violet (UV)-visible spectroscopy

nuclear factor kappa B (NFkB)

apolipoprotein E knockout (APOE-/-)

proprotein convertase subtilisin/kexin type 9 (PCSK9)

phosphate-buffered saline (PBS)

 ethylenediaminetetraacetic acid (EDTA) 

5- Use appropriate abbreviations (e.g. h for hours...)

Authors: we changed hours for h

Reviewer #3: Congratulations to all authors involved in this important manuscript entitled "Change in lipoproteins associated with pharmacological strategies in acute myocardial infarction”

Just to make it more understandable to readers, I suggest minor changes to the text.

1- I suggest standardizing the term (STEMI) to “ST-segment elevation myocardial infarction”, Instead of "ST-segment elevation acute myocardial infarction) - line 28

Authors: we changed as suggested. (see page 2, line 31,32)

lipid-lowering strategies, commonly prescribed in subjects with ST-segment elevation myocardial infarction (STEMI), on the pattern of LDL particles assessed by complementary methods.

2- Patients who were unsuccessful after thrombolysis and were referred for rescue angioplasty were included in this cohort?

Authors: few patients were excluded due to clinical instability (n=4). However, some stable patients showed occluded coronary arteries, and others despite non-reperfusion criteria, their arteries were not occluded on coronary angiography. 

3- Regarding the comparisons between the groups, using the non-parametric test, I suggest some considerations for a better understanding of the data

The Kruskal–Wallis test by ranks, Kruskal–Wallis H test is a non-parametric method for testing whether samples originate from the same distribution. It is used for comparing two or more independent samples of equal or different sample sizes. It extends the Mann–Whitney U test, which is used for comparing only two groups. The parametric equivalent of the Kruskal–Wallis test is the one-way analysis of variance (ANOVA). The test does not identify where this stochastic dominance occurs or for how many pairs of groups stochastic dominance obtains. For analyzing the specific sample pairs for stochastic dominance, Dunn's test, pairwise Mann–Whitney tests with Bonferroni correction.

Thus, in this manuscript, I did not observe the new alpha value, post Bonferroni correction (original alpha divided by the number of comparisons). In this case 0.05/4, which will be the post-test alpha, to accept or reject the null hypothesis (equality).

Authors: as was suggest by other reviewer, we added a paragraph in the statistical analysis. (see page 7, lines 4-6). 

Authors: for the quality of LDL analysis, we compared T0 with T6 using non-parametric test, as these data did not have Gaussian distribution. Statistical analysis was reviewed and we chose Dunn’s test after a Kruskal-Wallis test statistically significant, to determine exactly which groups are different. This analysis seems appropriate. (Reference: Dinno, A., 2015. Nonparametric pairwise multiple comparisons in independent groups using Dunn's test. The Stata Journal, 15(1), pp.292-300. 

6. PLOS authors have the option to publish the peer review history of their article. If published, this will include your full peer review and any attached files.

Authors: Agree.

---

## [Decision Letter · Decision Letter 1]

8 Aug 2022

Changes in lipoproteins associated with lipid-lowering and antiplatelet strategies in patients with acute myocardial infarction

PONE-D-21-17809R1

Dear Dr. Fonseca,

We’re pleased to inform you that your manuscript has been judged scientifically suitable for publication and will be formally accepted for publication once it meets all outstanding technical requirements.

Kind regards,

Salvatore De Rosa

Academic Editor

PLOS ONE

Additional Editor Comments (optional):

Reviewers' comments:

Reviewer's Responses to Questions

**Comments to the Author**

1. If the authors have adequately addressed your comments raised in a previous round of review and you feel that this manuscript is now acceptable for publication, you may indicate that here to bypass the “Comments to the Author” section, enter your conflict of interest statement in the “Confidential to Editor” section, and submit your "Accept" recommendation.

Reviewer #1: All comments have been addressed

Reviewer #2: All comments have been addressed

2. Is the manuscript technically sound, and do the data support the conclusions?

Reviewer #1: (No Response)

Reviewer #2: Yes

3. Has the statistical analysis been performed appropriately and rigorously? 

Reviewer #1: (No Response)

Reviewer #2: Yes

4. Have the authors made all data underlying the findings in their manuscript fully available?

Reviewer #1: (No Response)

Reviewer #2: Yes

5. Is the manuscript presented in an intelligible fashion and written in standard English?

Reviewer #1: (No Response)

Reviewer #2: Yes

6. Review Comments to the Author

Reviewer #1: (No Response)

Reviewer #2: Pape r titled (Changes in lipoproteins associated with lipid-lowering and antiplatelet strategies in patients with acute myocardial infarction), thanks for addressing the reviewer's comments

7. PLOS authors have the option to publish the peer review history of their article (what does this mean?). If published, this will include your full peer review and any attached files.

Reviewer #1: No

Reviewer #2: **Yes: **Sawsan Zaitone

---

## [Editor Report · Acceptance letter]

18 Aug 2022

PONE-D-21-17809R1 

Changes in lipoproteins associated with lipid-lowering and antiplatelet strategies in patients with acute myocardial infarction 

Dear Dr. Fonseca:

I'm pleased to inform you that your manuscript has been deemed suitable for publication in PLOS ONE. Congratulations! Your manuscript is now with our production department. 

Kind regards, 

on behalf of

Dr. Salvatore De Rosa 

Academic Editor

PLOS ONE